# A Comparison of Dry Period Outcomes after Selective Dry Cow Therapy Carried Out by Farm Staff versus Veterinary Students in a Low-Cell-Count Dairy Herd

**DOI:** 10.3390/ani13142318

**Published:** 2023-07-15

**Authors:** Peter Plate, Steven van Winden

**Affiliations:** Farm Animal Health and Production Group, Department for Pathobiology and Population Sciences, Royal Veterinary College, Hawkshead Lane, Hatfield, Hertfordshire AL9 7TA, UK

**Keywords:** selective dry cow therapy, mastitis, student teaching, drying off routine, somatic cell counts (SCC)

## Abstract

**Simple Summary:**

Selective dry cow therapy involves giving local antibiotic treatment into the udder only to those cows which are likely to be infected with a mastitis-causing organism, while uninfected cows receive an internal teat sealant only, a physical barrier inserted into the teat to prevent infections entering the udder via the teat canal. The application of teat sealant only without antibiotics can induce infections into the udder if not done hygienically, and veterinarians should be training farmers in hygienic application. Therefore, veterinary students have to be trained in the technique themselves, and farmers may be concerned about allowing students to practice on their cows. This study follows up cows dried off by farm staff and students on a single dairy farm and compares several parameters indicative of mastitis and also looks into the survival of cows in the herd. There is no indication that cows perform worse when dried off by students under close supervision, and the risk of culling within twelve months was lower in cows dried off by students.

**Abstract:**

(1) Background: Selective dry cow therapy is widely promoted in many countries worldwide, however, concerns have been raised about the consequences of the unhygienic application of preparations by untrained operators, especially if no antimicrobials are being used, risking deteriorating mastitis outcomes. (2) Method: This study follows up on cows being dried off by farm staff and those dried off by final-year veterinary students and first-year graduate interns in a supervised training session. Subsequent mastitis parameters and culling data in a single herd with a low somatic cell count were evaluated. (3) Results: A total of 316 dry periods were enrolled in the study. There was no significant difference in the percentage of cows showing at least one high SCC reading within 90 days of the following lactation or cows with at least one case of clinical mastitis within the same period, neither in the total nor in the subset of cows dried off without an antimicrobial. Dry period cure rates and dry period new infection rates were similar too, as was the percentage of cows surviving in the herd after six months. The risk of culling within twelve months post-drying off was lower in cows dried off by students, the difference in survival manifesting itself from 150 days post-drying off, which is an unexplained finding. (4) Conclusion: Well-supervised practical training sessions on drying off routine can be responsibly implemented on well-managed commercial dairy herds.

## 1. Introduction

The dry period is a crucial part in the cow’s lactation cycle with regards to reducing intramammary infections (IMIs). Natural defense mechanisms, like forming a keratin plug in the teat canal in order to prevent pathogen entry or lactoferrin to deplete bacteria of iron [1], play a crucial role in reducing clinical and subclinical mastitis. Dry cow therapy aims to support these defenses, therefore leading to a reduction in new IMIs as well as curing any existing ones. Until around 2000, both preventive and therapeutic aims were achieved by blanket antibiotic dry cow therapy (BDCT), which involved introducing a long-acting antibiotic intramammary suspension into each quarter. This was part of the five-point plan established in the 1960s, which was very successful in reducing the incidence and prevalence of contagious mastitis, predominantly caused by *Staphylococcus aureus* and *Streptococcus agalactiae* [2].

The successful reduction of contagious mastitis led to a decrease in the number of cows infected at the point of drying off. This and the introduction of internal teat sealants (ITSs) led to the concept of selective dry cow therapy (SDCT) [3]. Internal teat sealants provide a mechanical barrier at the teat and have been shown to be more effective in preventing new intramammary infections than antimicrobials [4]. It is known that a percentage of cows fail to produce an effective keratin plug, more so in higher-yielding cows [5]. It follows that individually selecting cows based on their infection status for either antibiotic dry cow therapy (ADCT) combined with an ITS, or ITS only, was a logical step forward, leading to potentially comparable outcomes with reduced antimicrobial usage. Kabera and others (2021) carried out a systematic review and meta-analysis comparing SDCT and BDCT and concluded that SDCT leads to a reduction in antimicrobial usage without any difference in the risk of acquiring new IMIs, curing existing IMIs and showing clinical mastitis within the first four months of calving, if ITSs are used in SDCT [6].

As for the selection criteria, somatic cell count and clinical mastitis records are normally used in the UK, and cows are commonly considered infected if any of the last three monthly SCC readings is above 200,000 or if a clinical case of mastitis occurred in the three months before drying off [7]. Higher or lower thresholds may be used dependent on herd status, affecting sensitivity, specificity and predictive values [8,9,10].

The current industry standard is to select at cow level, therefore applying ADCT to all four quarters in cows judged as infected. Treatment selection at the quarter level, only applying ADCT to infected quarters, based on quick tests like California Mastitis Test or on farm culture tests have been described with encouraging results [11,12,13,14].

Few studies have assessed direct effects of antibiotic versus selective dry cow therapy on antimicrobial resistance and focus mainly on residues in colostrum and effects in calves, with contradictory results [15,16]. In the wider context of a One Health approach, however, it is necessary to minimise any unnecessary use of antimicrobials [17], and EU regulation 2019/6, article 107, prohibits the routine use of prophylactic antimicrobial usage.

In the United Kingdom (UK), despite the promotion of selective dry cow therapy by veterinarians and milk buyers, sales of antimicrobial dry cow intramammary products have fallen by only 12% between 2014 and 2021, from 0.62 to 0.55 courses per cow per year. In comparison, sales of lactating cow intramammary products, used to treat clinical mastitis, have fallen by 55% in the same time period [18]. While the latter figure was also confounded by supply issues regarding lactating cow products, it is widely believed that there is more potential to promote and implement SDCT in UK dairy herds. There are, however, known obstacles preventing the wider uptake of SDCT.

Ref. [19] investigated barriers and facilitators for SDCT on Irish dairy farms by telephone interviews and identified fear of increasing mastitis, infrastructure limitations, lack of preventive advice and peer influence as barriers, while targeted veterinary consultancy, regulatory pressure and high standards of farm hygiene were facilitators. Farmers’ attitudes in the Netherlands were examined by Scherpenzeel and others (2016) [20], and obstacles were lengthy experience with BDCT and resistance to change, fear of introducing pathogens and iatrogenic mastitis and even deaths, and skepticism about udder health outcomes using fewer antimicrobials. Scherpenzeel and others (2018) [21] also investigated vets’ attitudes to SDCT and found that some vets were negative about the approach, leading to less implementation of SDCT. In a survey in the United Kingdom, Orpin (2017) [22] found that 55% of farmers surveyed feared that selective dry cow therapy could result in more deaths and mastitis. Ref. [23] described a case of gangrenous mastitis as part of a herd outbreak in a dairy herd following combination dry cow therapy with cephalonium and a bismuth-based teat sealant. *Pseudomonas aeruginosa* was cultured from the udder secretion as well as from the teat sealant tube which was stored in an open bucket in the parlour, with the bottom layer of tubes immersed in contaminated water. Cephalonium is ineffective against *Pseudomonas* sp.

It is well established that veterinary input and consultancy drive farmers’ confidence and success in implementing SDCT [19]. However, this requires veterinarians to be trained in the theory and practice of using the technique and applying a hygienic routine. Knowing the potential risks and perceived barriers, a confident approach is needed to implement this approach, which also requires practical confidence amongst the training vets themselves in the method of drying off. Edgar Dale’s Pyramid of Learning compares retention rates of different learning methods, postulating that the highest retention rates occur after active learning like “practice by doing” and “teaching others” and the lowest ones occur after passive learning like listening to lectures, reading and audio-visual exposure [24]. Although the concept and the actual retention rate figures are disputed by some authors [25], the general concept is widely agreed and is also similar to other theories like deep and surface learning [26] and multisensory learning [27]. Reference [28] examined veterinary practical classes on their deep learning scores and found higher scores in practical sessions compared to more passive methods. Overall, while lectures, audio-visual media and models provide a valuable preparation of students and farmers to learn the technique of SDCT, concerns may arise on mastitis outcomes in those cows used for teaching. In the UK, practical teaching of drying off procedures on live cows is not routinely taught to veterinary students, but the routine is explained and some teaching takes place using models.

In order to assess the position that well-trained novices to a new technique are as good as experienced practitioners of said technique, this study follows cows from a single dairy farm where the practical teaching of veterinary students and recent graduates (interns) was carried out and compares clinical and subclinical mastitis outcome parameters of cows dried off by students versus those dried off by farm personnel.

It is hypothesised that teaching well-supervised students in drying off cows using SDCT does not lead to negative mastitis outcome or longevity parameters in the cows compared to cows dried off by experienced farm staff.

## 2. Materials and Methods

### 2.1. Farm Details and Background

The 150-cow Holstein herd is part of an agriculture college in the southwest of England. Apart from involving students and apprentices in some activities, it is run under commercial conditions. During the study period, two herdspeople, each with more than 10 years’ experience, were responsible for the practical cow management including drying off cows. The herd is predominantly autumn calving; most cows calve between August and October, with a few cows calving up to December. Drying off in the autumn block takes place mainly from June until September, with a few cows dried off in October. After drying off, cows are turned out to pasture for about five weeks before being housed for a three-week transition period pre-calving. Yields per cow per year increased from about 9000 to 10,000 litres during the study period. Following a historic fertility problem, a small spring calving group of about 15 to 25 cows was established in recent years, calving in March and April. Cows are kept in a cubicle barn during the winter, with mattrasses and sawdust as the lying surface. They are fed on a partial mixed ration containing maize silage, grass silage and a protein blend, with compound feed offered in the milking parlour according to individual yields. During the summer, cows are grazing pastures and fed compound cake at milking in the parlour. Dry cows for the autumn calving block are first kept on pasture and are housed in two designated dry cow barns with loose housing on straw for the transition period. Those are fed a total mixed ration containing maize silage, grass silage, protein blend and a high amount of straw. The dry period length is 65–70 days.

The overall mastitis incidence during the study period is around 25 cases per 100 cows per year, somatic cell counts average at around 135,000/mL. The herd replacement rate is around 29%.

### 2.2. Student Training Sessions

Between 2019 and 2022, practical teaching sessions on drying off cows were carried out for a proportion of the autumn calving blocks during the summer months. These were performed on given teaching days on those cows due to be dried off at that time, making them a convenience sample. The trainees were final-year veterinary students on a rotation and first-year qualified veterinarians on an internship programme. Most students and interns on the course had never dried off cows before. The practical sessions in the parlour were preceded by a seminar on selective dry cow therapy and a practical session on a model udder.Between two and four students took part in each training session, with six to ten cows dried off on a single day, so most individual students dried off between two and four cows. One cow was dried off at a time, and all students were closely supervised by one of the authors (PP) during the whole of the procedure.

The following methods were taught, based on UK industry recommendations [8,29], as follows:Cows to be dried off were separated at morning milking to re-enter the washed-down herringbone parlour afterwards.All tubes, cotton wool and spirit were stored in closed containers or packaging, avoiding contamination during the process.Cows had been allocated by the farmer to either antibiotic or non-antibiotic treatment according to the criteria in the health plan set up with the local veterinarian, and a list was provided.New nitryl gloves were worn by all students which were either washed with warm water or changed between cows.A pre-milking teat disinfectant was applied via a dip cup to all four teats and kept on for at least 30 s, then wiped off with a clean paper towel, one towel per cow.Teats were wiped with cotton wool swabs soaked in surgical spirit, with new swabs used until no soiling of the cotton wool was visible, but with a minimum of two swabs per teat. One hand was holding the base of the teat, the other applying the swab in a rotating movement under gentle pressure, concentrating on the area of the teat end. To avoid subsequent contamination, wiping was started with the teats away from the operator, followed by the wiping of the near teats. Milk was then stripped out a couple of times to visualise the teat orifice.In those cows due to receive an antimicrobial, one tube containing cephalonium (Cepravin Dry Cow^TM^) was infused into every quarter, using the partial insertion technique. This started at the teats closest to the operator, followed by those far away. Once infusion was complete the quarter was massaged gently, facilitating diffusion of the product. Teats were wiped with a single swab per teat before applying the teat sealant.All cows received an internal teat sealant containing bismuth subnitrate (Orbeseal^TM^), either as a standalone treatment or after infusion of the antimicrobial product. This started at the teats nearest to the operator and again was done using the partial insertion technique, inserting the tip of the nozzle only to avoid teat damage. During the infusion the base of the teat was “clamped” using the thumb and index or middle finger in order to retain the sealant within the teat.After insertion, a post-milking teat disinfectant was applied using a spray.All dried off cows were marked with three red tapes around the tail.After drying off the cows were moved through a race and crush where they received an oral liver fluke treatment containing triclabendazole, before being moved onto a pasture. They had no access to a lying area for at least 30 min.Cow number, date, treatments given and the name of the student were all recorded and later entered into an Excel^TM^ spreadsheet.

The same drying-off protocol was also applied by farm staff, as outlined in the farm’s health plan which was drawn up in conjunction with the routine veterinarian.

### 2.3. Criteria for Selective Dry Cow Therapy

The criteria for the use of antimicrobials at drying off were the same for cows dried off by students and farm staff and were as follows according to the health plan: any of the preceding three-monthly milk recordings with a cell count over 150,000/mL OR any clinical mastitis case within three months before drying off.

### 2.4. Data Management and Statistical Evaluation

For further information, animal records were obtained using Interherd plus which accesses the farmer’s milk recording data, and Uniform agri, a desktop herd management programme on the farm, where all treatments are recorded.

All cows not recorded as being dried off by students were dried off by farm staff, and no other students were involved in drying off cows. All cows dried off to calve in the autumn calving block were enrolled in the study. Cows from the small spring calving blocks were excluded, as no student training sessions were held during their drying off.

In order to compare the students’ performance with that of farm staff the following outcomes were compared:-Somatic cell counts post-calving

Cow records were assessed to check whether any of the monthly milk recordings within 90 days of the following calving revealed at least one SCC reading above 200,000/mL. Cows were milk recorded from five days in milk. Somatic cell count data was collected by National Milk Records using an automated analyzer.

-Dry period new infection rates

Of those cows dried off with a last SCC reading of below 200,000/mL before drying off, the percentage of cows with a first reading of above 200,000/mL post calving is defined as dry period new infection rate and compared between the cohorts.

-Dry period cure rates

Of those cows dried off with a last SCC reading of above 200,000/mL before drying off, the percentage of cows with a first reading of below 200,000/mL post-calving is defined as dry period cure rate and compared between the cohorts.

-Clinical mastitis post-calving

Cow records were assessed whether cows developed at least one case of clinical mastitis in the 90 days following parturition. Cows culled before 90 days and not showing a clinical case up to then were excluded from the analysis.

-Survival in the herd

Cow records were assessed to check whether cows remained in the herd six and twelve months following the drying off procedure.

To compare SCC and clinical mastitis in the first 90 days post-calving and six- and twelve-month survival in the herd, a multivariable analysis was carried out. Variables in the equation are: lactation number, co-treatment with antimicrobials (yes/no), student/farm staff dry-off and the interaction between the latter two. The significance level was set at a *p*-value of 0.05. All comparisons except for dry period cure and new infection rates and culling analyses were carried out on the total of enrolled cows as well as on the subset of cows dried off without antimicrobials only, as the main hygienic concerns are directed at the latter. All statistical analyses were carried out using SPSS (IBM).

## 3. Results

### 3.1. Dry Periods Enrolled and Test for Bias

A total of 316 dry periods were enrolled during the four years from 2019 to 2022. In total, 118 dry periods involved first lactation cows, 89 second lactation cows, 67 third lactation cows and 42 fourth or more lactation cows. In 134 events cows were dried off by students and in 182 events cows were dried off by farm staff. Of the 316 drying off procedures, 229 involved no antimicrobials (72.5%), 77 included an antimicrobial (24.4%), and in 10 drying off events the treatment was unknown or the records contradictory (3.1%).

As the allocation to the groups was not randomised but based on the teaching schedule, cows dried off by students and farmers were compared for potential bias regarding two factors: primiparous versus multiparous cows and cows with and without antimicrobial treatments. Chi square tests revealed *p*-values of 0.34 for primiparous versus multiparous cows and 0.07 for antibiotics given vs. no antibiotics given, so there is no bias in the lactation numbers but a trend without significant difference in students having dried off more cows with antibiotics than farm staff (30% versus 21%).

### 3.2. Results of the Multivariable Analysis

Total dry periods

Somatic Cell Counts in the first 90 days post calving

A total of 305 dry periods with sufficient data were included in the analysis. Of the 175 cows dried off by farm staff, 23% (41 cows) developed at least one SCC reading above 200,000/mL, and of those dried off by students this number was 25% (32 out of 130). The difference is not significant (*p* = 0.648).

Clinical mastitis in the first 90 days post calving

Cows not developing a case of mastitis were only included if they were at least 90 days in the herd post-calving. A total of 300 dry periods with sufficient data were included in the analysis. Of the 171 cows dried off by farm staff, 24 cows (14%) developed at least one case of clinical mastitis within 90 days of the subsequent lactation, and of those dried off by students the figure was 13 out of 129 (10%). The difference is not significant (*p* = 0.373).

Dry period cure rates—last SCC highs only, compared with first SCC

A total of 36 cows (18 dried off by farmers and 18 by students) had a high SCC pre-drying off and sufficient data to enter the dry period cure rate analysis. Based on the first SCC reading in the following lactation (within 40 days of calving), dry period cure rates were 61% for students and 83% for farm staff (*p* = 0.212)

Dry period new infection rates—last SCC lows only, compared with first SCC

A total of 252 cows with a low cell count pre-drying off had sufficient data post-calving to enter the analysis: 158 by farmers and 94 by students. Dry period new infection rates based on the first SCC reading post-calving (within maximum of 40 days post calving) were 17% for students and 14% for farm staff (*p* = 0.313).

Culling analysis

The six- and twelve-month survival rates were established from dry periods carried out at least 12 months before the analysis. Of the 118 eligible cows dried off by students, 110 were still in the herd six months later (93%), and of the 132 dried off by farm staff 120 (91%) were still in the herd after six months (*p* = 0.377). After drying off, 90% of the 118 eligible cows dried off by students were still in the herd after 12 months, while of the 132 dried off by farm staff the figure was 83% (*p* < 0.001). The Kaplan–Meier survival curve over 12 months shows a similar survivability of cows in the first 150 days (dry period plus three months’ lactation) but a difference between 150 and 365 days (Figure 1).

Dry periods without antibiotic treatment

Somatic Cell Counts in the first 90 days post calving (non-antibiotic dry periods only)

In total, 223 dry periods entered the analysis, 91 of which were dried off by students. Of the 132 cows dried off by farm staff, 21% (28 cows) developed at least one SCC reading above 200,000/mL, while of those dried off by students this number was 19% (17 out of 91). The difference is not significant (*p* = 0.857).

Mastitis in first 90 days (non-antibiotic dry periods only)

A total of 220 dry periods with sufficient data were included in the analysis. Of the 130 cows dried off by farm staff, 15 cows (12%) developed at least one case of clinical mastitis within 90 days of the subsequent lactation, while of those dried off by students the figure was 7 out of 90 (8%). The difference is not significant (*p* = 0.284).

The results of the multivariable analysis are summarised in Table 1.

In summary, no significant differences were found in the mastitis outcome parameters:At least one high SCC within 90 days of the following lactation;At least one case of clinical masitis within 90 days of the following lactation;Dry period cure rate;Dry period new infection rate;Survival in the herd after six months.

Significant effects of students vs. farm staff could be observed for the twelve-month survival. Additionally, in the multivariable analysis, lactation number was a significant factor on SCC post-calving, mastitis post-calving, dry period new infection and 365-day survival rate and risk, but not dry period cure rate. Figure 1 shows the Kaplan–Meier survival curve for the twelve-month post-drying off period.

## 4. Discussion

This is the first study to the authors’ knowledge comparing mastitis outcomes in cows dried off by veterinary students/interns and farm staff. This could deflect commercial dairy farmers raising concerns about allowing student training sessions taking place on their enterprises, seeing potential risks in unhygienic application of treatments and the subsequent increase of new IMIs or a failure to cure existing ones. These concerns may be greater in herds with low SCCs and low clinical mastitis incidence, as any failures to apply the correct hygiene may manifest itself in the deterioration of udder health. The current results are encouraging as they have been obtained on a herd with few mastitis problems. In addition, farmers in high SCC/high clinical mastitis herds may be reluctant to allow students to train on their cows, as every effort is to be directed towards improving the situation. The drying-off regime is only one amongst several factors determining dry period performance. Preparing cows for drying off (controlling nutrition, body condition and milk yield), dry cow housing and calving management are other important aspects, studies on which have been well published within the UK dairy industry [29]. As the comparison was carried out within one farm, the management factors can be considered as constant.

We focussed on the mastitis outcomes in the first 90 days post-calving, which is in line with Green and others (2002) [30] and Bradley and Green (2004) [1], who concluded that this period reflects the clinical mastitis cases originating from dry period intramammary infections. If there was a difference in clinical mastitis incidence during this period, the drying-off routine would be a potential factor, as all other management factors during the dry period and early lactation were the same for both cohorts of cows. On a SCC level (subclinical mastitis), the three-month period is less established in the literature, therefore both the single first SCC reading post-calving (if within 40 days post-calving) was taken into account to calculate dry period cure rates and dry period new infection rates, as well as all SCC readings in the first 90 days.

Clinical mastitis is a welfare issue at any stage of lactation, as it is associated with pain and an alteration of the pain threshold, even in mild cases [31]. Mastitis in early lactation, attributed to dry period infections, leads to higher economic losses. Rollin and others (2015) [32] estimated the cost of an average case of clinical mastitis within 30 days of calving to be USD 444, predominantly due to indirect costs like future milk production loss and higher risk of premature culling. Lucey and Rowlands (1984) [33] established a higher subsequent milk loss after mastitis cases early in lactation compared to those in late lactation.

The allocation of cows to students and farmers was not made randomly but was based on a convenience sample of cows due to be dried off by students at given teaching days. While there was no question of bias regarding primiparous vs. multiparous cows there was a trend just above the 0.05 significance level of students drying off a higher percentage of cows with antibiotics, so the students appeared to have dried off a higher proportion of potential problem cows.

The data show that cows dried off by students in a training session performed equally to those dried off by experienced farm staff in most areas examined, that is SCC and clinical mastitis within 90 days of the following lactation, dry period cure and new infection rates as well as survivability in the herd up to six months post-drying off. There was no difference in survival in early lactation, as both the Kaplan–Meier survival curve and the six-month survival rates show, so no difference in deaths or culling soon after drying off could be observed. However, the difference in survival at 12 months is a surprising and unexplained finding. As no culling reasons were available, the cause of this difference remains unclear. If mastitis parameters played a part, the fact that in the student training sessions no restrictions were placed on the available time for the procedure and the amount of materials used, in combination with a natural fear by students of deteriorating a cow’s condition when performing a task for the first time, this may have led to an improved routine compared to what is practically achievable in a routine farm situation with the associated pressures on labour. This aspect requires further investigation, as there may be potential for improvement even in a low-problem herd with experienced staff.

The data show that well-supervised training sessions by students on well-managed commercial farms do not negatively affect the welfare and economic performance of the cows. However, as a limitation, it has to be pointed out that this result was achieved on one farm only, and outcomes may differ on other farms or in training sessions performed in different ways.

The effectiveness of teaching in live cows has been underlined in other areas such as teaching pregnancy diagnosis—Annandale and others (2018) [34] found that students trained on live cows showed a higher sensitivity to detect pregnancies under 6 months by transrectal palpation than those trained on a model. However, the value of models as a precursor for teaching on live cows is well established, e.g., by Giese and others (2016) [35]. There is some reluctance amongst farmers and teachers that using live cows in a commercial situation may impact welfare or economic outcomes. As for the teaching of drying off practice the current study provides encouraging results in favour of “real life” teaching. While the inexperience of the students may be seen as detrimental, an advantage of the teaching sessions is the lack of a time constraint. While labour is under pressure in dairy farming to carry out tasks quickly, the teaching sessions had no limit on time, therefore allowing students to carry out the routine to the highest standards while being closely observed and advised by an experienced educator.

## 5. Conclusions

Teaching future farm veterinarians and advisors on a crucial procedure such as selective dry cow therapy has the potential to promote the concept and avoid negative effects due to poor hygiene at the application. The data show that carrying out well-supervised training sessions on well-managed commercial dairy farms, allowing sufficient time for students, is a responsible and effective way of teaching with no detectable impacts on animal welfare and farm economics.

## Figures and Tables

**Figure 1 animals-13-02318-f001:**
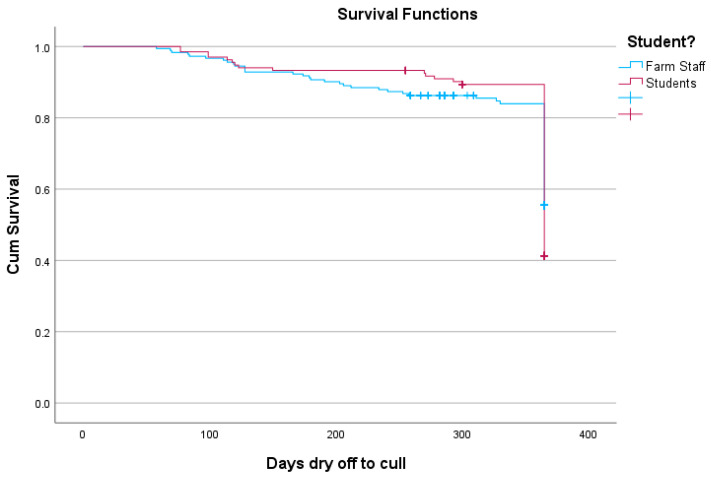
Kaplan–Meier survival curve for the twelve months post-drying off, comparing farm staff and students in a single herd in the UK.

**Table 1 animals-13-02318-t001:** Summary of the results of the multivariable analysis, reporting the contrast between farm staff vs. recently trained students, controlled for lactation number and co-treated with antimicrobial, in a single herd in the UK (SCC = somatic cell count).

Parameter	Farm Staff	Students	*p*-Value
Overall			
% with at least one SCC over 200,000/mL within 90 days post-partum	23 (41/175)	25 (32/130)	0.648
% with clinical mastitis within 90 days post-partum	14 (24/171)	10 (13/129)	0.373
Dry period cure rates	83 (15/18)	61 (11/18)	0.212
Dry period new infection rates	14 (22/158)	17 (16/94)	0.313
% of cows in herd six months after drying off	91 (120/132)	93 (110/118)	0.377
% of cows in herd twelve months after drying off	83 (110/132)	90 (106/118)	<0.001
Cows dried off without antimicrobials:			
% with at least one SCC over 200,000/mL within 90 days post-partum	21 (28/132)	19 (17/91)	0.857
% with clinical mastitis within 90 days post-partum	12 (15/130)	8 (7/90)	0.284

## Data Availability

The farmer gave consent for the raw data to be used by the researchers only, but any queries involving the processing of these data can be directed to the corresponding author.

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
