# Peer review of "A Comparison of Dry Period Outcomes after Selective Dry Cow Therapy Carried Out by Farm Staff versus Veterinary Students in a Low-Cell-Count Dairy Herd"

_animals, 2023, doi:10.3390/ani13142318_

Round 1
Reviewer 1 Report
The article covers very important aspects of mastitis prevention, as well as significant topic of vet students practical trainings and education of farmers/farm's staff. The number of dried periods analyzed is sufficient for single herd.
Please add SCC to the key words list
The reviewer would advice adding to introduction and/or discussion broader aspects of OneHealth/EU action towards reduction of antibiotic usage & problems with resistance.
What is the common time/% of "hands on" trainings during vet studies in UK?
Please give more details about the cows & staff in M&M part (later e.g. primiparous and multiparous cows are compared). SCC - instrumental method used for counting?
Paragraph 2.2 presenting well described details of training sessions routines
Which p-values were taken as significant? In the M&M part please complete the statistical analysis part by pointing the tests used.
Discussion is quite brief, but touches most important aspects related to the results.
Big part of references list is based on actual literature.
As we need to be careful with ethical issues the reviewer advices to add information to the article that supervisors were concerned about cows welfare during the trainings.
Reviewer 2 Report
REVIEW for the journal Animals (ISSN 2076-2615)
Article “A comparison of dry period outcomes after selective dry cow therapy carried out by farm staff versus veterinary students in a low cell count dairy herd”
Manuscript ID: animals-2465428
Authors: Peter Plate, Steven Van Winden
Brief summary. The dry period plays an important role in the cow's lactation cycle in reducing intramammary infections, and input from veterinary professionals and farmer consultation during this period is critical to successful herd health improvement. This study follows cows from a single dairy farm where practical teaching of veterinary students and recent graduates (interns) was carried out, and compares clinical and subclinical mastitis outcome parameters of cows dried off by students versus those dried off by farm personnel.
Considering this, the article can be relevant to university staff organizing student internships as well as farm consultants assisting in practical solutions to the mentioned problem.
General concept comments
1. Introduction. Although the authors provide a detailed and coherent description of the importance of the dry period in cows and methods for reducing mastitis in dairy herds, at the end of the literature review, they should clearly formulate the hypothesis and objectives of the article, providing scientific value and content to it.
2. Materials and Methods. The student training sessions are described in great detail, which is undoubtedly important for students and their internship organizers, but I do not see any scientific content in it.
3. Data management and statistical evaluation. I missed a specific description of the statistical methods used, taking into account the hypothesis and objectives of the article.
4. The description of the results should be more detailed.
Specific comments
1. What statistical software was used for the statistical data analysis?
2. What statistical tests were used and for what purposes?
3. All statistical indicators presented in the first table should be described in the methodology, and their abbreviations should be explained.
4. Lines 256-257: 95CI? Perhaps it means 95% CI (confidence interval).
5. The results of Figure 1 should be described in detail.
Conclusion. The authors' data, which indicate that conducting well-supervised training sessions on well-managed commercial dairy farms, with adequate time allocated for students, is a responsible and effective approach to teaching that does not have a negative impact on animal welfare and farm economics, in my opinion, are suitable for publication in a practical journal but lack scientific novelty and content.
Sincerely, reviewer.
Reviewer 3 Report
This study presents results of a comparison between selective dry-cow treatments performed by farm personnal vs closely supervised students in 1 herd with good udder health in UK. The topic is of interest but only for teaching veterinary institutions. It was performed in only 1 herd so the generalisation of the results should be made with caution.
Specific comments:
Line 3. Carried
Lines 25, 240, 241. This reviewer suggest to replace periods by cows since periods are not usually enrolled.
Introduction. Please add reference for several sentences. lines 42, 50, 51.
Line 51. Do not start a sentence using an abbreviation.
Line 91. SDCT?
Line 122. Should begin the material and method section with an ethical approval number for this project.
Lines 140-147. How many different teachers? How many trainees per teachers? Define closely supervised. One on one? How many different trainees? How many cows per trainees? Only one cow? Only one teat? Please give more details.
Line 146. Preceeded?
Lines 166-168. Stripping was done after wiping? If yes, this can cause recontamination of the teats with milk or by milkers. What is the goal of wiping far teats before near teats if the stripping happens afterward? Unclear.
Line 184. Cows were kept on pasture until next calving? Explain the transition period procedures on this farm. Any mastitis vaccines used?
Line 188. How many different farm personnel did dry-cow treatments? You state that they were well trained personnel. For how many years they were doing it? Give more details on farm personnel please.
Lines 206-208. Please define when SCC testing is starting after calving. 5 DIM?
Table 1 and figure 1. Add some info in the title. Should be interpreted stand alone so at least say that is is a study about dry-cow treatment in a herd in UK.
Table 1. Abbreviations used in the table but not defined . Please avoid abbreviations or define them. (SCC, pp)
Line 263. vs
Line 271. Post.
Lines 347-350. The study was performed only on 1 dairy farm. You didn't prove that it would be the case in several herds with different procedures or settings. Also, you did not measure animal welfare or farm economics in this study. Maybe having students in the parlour manipulating the cows longer than usual is a very stressful event for these cows and welfare could be impaired. Welfare is not related just to clinical mastitis cases. You need to keep the conclusions related to your objectives and your study design. Please rephrase.
Lines 356-358. An ethical approval is required for the use of animals for teaching purpose in all the institutions I know. Even if the data were used retrospectively, an ethical approval was needed by definition even before the study was done.
Round 2
Reviewer 2 Report
The authors have taken into account the provided comments and significantly improved the quality of the article, therefore my suggestion to the journal editorial board is as follows: accept in present form.